# Recurrent Selection with Low Herbicide Rates and Salt Stress Decrease Sensitivity of *Echinochloa colona* to Imidazolinone

**Lariza Benedetti [1], Vívian Ebeling Viana [1] , Pâmela Carvalho-Moore [2] , Vinicios Rafael Gehrke [1] , Gustavo Maia Souza [3], Edinalvo Rabaioli Camargo [1] , Luis Antonio de Avila [1,* ] and Nilda Roma-Burgos [2,* ]**

[1] Department of Crop Protection, Federal University of Pelotas, Pelotas 96160-000, RS, Brazil; larizabenedetti13@hotmail.com (L.B.); vih.viana@gmail.com (V.E.V.); viniciosraffael@hotmail.com (V.R.G.); edinalvo_camargo@yahoo.com.br (E.R.C.)

[2] Department of Crop, Soil and Environmental Sciences, University of Arkansas, Fayetteville, AR 72704, USA; pcarvalh@email.uark.edu

[3] Department of Botany, Federal University of Pelotas, Pelotas 96160-000, RS, Brazil; gumaia.gms@gmail.com

* Correspondence: laavilabr@gmail.com (L.A.d.A.); nburgos@uark.edu (N.R.-B.)

**Abstract:** Weeds represent an increasing challenge for crop systems since they have evolved adaptability to adverse environmental conditions, such as salinity stress. Herbicide effectiveness can be altered by the quality of water in which the weed is growing. This research aimed to study the combined effect of salt stress and recurrent selection with a sublethal dose of imidazolinone herbicides in the shifting of the sensitivity of *Echinochloa colona* (L.) Link (junglerice) to imidazolinone herbicides. This study was divided into two experiments; in experiment I, three recurrent selection cycles were conducted in Pelotas/RS/Brazil with imazapic + imazapyr at $0.125\times$ the field rate; and in experiment II, three recurrent selection cycles were conducted in Fayetteville/AR/USA with imazethapyr, at $0.125\times$ the recommended dose. Salt stress was implemented by irrigation with 120 mM sodium chloride (NaCl) solution. The effective dose for 50% control of the population ($ED_{50}$) values increased from the field population to the second generation (G2) after recurrent selection with a sublethal dose of imidazolinone combined with salt stress, supporting the hypothesis of reduced susceptibility by the combination of these abiotic factors. Recurrent exposure to a sublethal dose of imazapic + imazapyr or imazethapyr, combined with salt stress, reduced susceptibility of *Echinochloa colona* (L.) plants to imidazolinone herbicides.

**Keywords:** adaptation; junglerice; low herbicide dose selection; reduced susceptibility; salinity

## 1. Introduction

Population projections indicate rapid and continued global growth in the coming decades, exceeding 9.5 billion people in 2050, increasing demand for food in quantity and quality [1]. The need to increase agricultural productivity directly reflects the emerging concern of population increase, as there are already restrictions on land use and several factors influencing production, including environmental conditions and management [2].

Weeds are one of the main problems causing damage and reduced rice yields worldwide, especially the weedy rice and *Echinochloa* species [3–5]. Selective grass control in rice became possible with the introduction of the Clearfield™ production system, starting in the early-2000s, with cultivars resistant to herbicides in the imidazolinone chemical group (e.g., imazamox, imazethapyr, imazapic, imazapyr) [6]. Nevertheless, grass weeds such as weedy rice, barnyardgrass, and junglerice quickly evolved resistance to these herbicides, reducing grain yield and creating socioeconomic problems [7–9].

There are currently 515 unique cases (species × site of action) of herbicide-resistant weeds globally, with 165 records, including *Echinochloa colona*, involving the acetolactate synthase (ALS) inhibitors [9]. Weed resistance to herbicides involves two main types

of mechanisms: (1) target-site resistance (TSR), which results from mutations or overexpression of the target enzyme; and (2) non-target-site resistance (NTSR), which involves mechanisms that minimize the amount of active herbicide reaching the targeted process in the plant, or physiological adaptations that protect the plant from the lethal effects of the herbicide [10]. While TSR is easily documented, NTSR is complex and not fully understood in many weed species. Either type of mechanism can confer resistance to multiple herbicides; both types of mechanisms can occur in the same population or the same plant, adding another layer of complexity [11].

Weed control has been almost exclusively done with herbicides and when used at the conditions recommended and at the registered label rate, they cause very high mortality [12,13]. However, 'low-dose selection' of weed populations occurs in the field all the time, for example, in crop production fields arises from insufficient coverage of some individuals partially covered by other plants; variations in per-plant dose due to differences in weed size, weed density, field topography, or soil type; drift rates to populations on field edges; and other biological, physical, or environmental factors [14–16]. The recurrent selection at sublethal doses and the dynamics of resistance evolution are being investigated by some research groups, as previously reported in *Amaranthus palmeri* with dicamba, *Avena fatua* with diclofop-methyl, *Lolium rigidum* with glyphosate, *Raphanus raphanistrum* with 2,4-D, among others [14,17–20].

We hypothesize that non-target site weed resistance evolution may be driven by sublethal rates of herbicides and by environmental stresses such as heat stress, drought stress, and salt stress [15,21]. Climate change impacts on the ecosystem need attention, such as soil and saline water management and the dynamics of interactions [22]. Soil salinity is a global problem that affects approximately 20% of cultivated land globally and 33% of irrigated land [23,24]. It is estimated that, by 2050, 50% of the arable land worldwide will be affected by salinity [25]. Soil salinity can reduce crop yields significantly, as reported in India and Pakistan, where the rice yield losses are 45% and 69%, respectively [26,27]. Salinity occurs naturally in the soil and water source material. Still, it can also be intensified by high temperature and evaporation, low rainfall and seawater intrusion, as well as human actions such as inadequate irrigation and drainage [23,28]. Salinity stress impacts many aspects of plant physiology, since it causes osmotic stress leading to a reduction of osmotic potential, nutritional imbalance due to high ion concentration, and inhibition of absorption of other cations, in addition to the toxic effect of sodium and chloride ions [29–31].

Weeds present an increasing challenge for crop production, since they have high genetic resilience and adaptation ability to adverse environmental conditions, including salinity stress [32]. Water quality and availability affect herbicide performance [33,34]. In this sense, there are assumptions of the crosstalk between genetic and epigenetic regulation, and the pathways involved in the abiotic stress and herbicide response in plants [35–37].

This research aimed to study the transgenerational effect of recurrent selection with combined salt stress and a sublethal dose of imidazolinone (imazapic + imazapyr or imazethapyr) herbicides on *E. colona*. We studied the effect of salt stress because, in some world regions, water or soil salinity is a problem in rice production fields.

## 2. Materials and Methods

*2.1. Experiment I—Effect of Salt Stress and Recurrent Selection With a Sublethal Dose of Imazapic + Imazapyr on the Herbicide Sensitivity of the E. colona Population from Southern Brazil*

2.1.1. Plant Material

This selection study was conducted using seeds of *E. colona* (referred to hereafter as G0), originally ECO-S (susceptible to imidazolinones), collected in 2014 from a field in Capão do Leão, RS, Brazil, with no previous history of imidazolinone herbicide application. This population (parental, G0) was subjected to three successive cycles of recurrent selection with a low dose of imazapic + imazapyr formulated mixture, with and without salt stress, to produce G1 and G2 progenies following the general procedure described below.

2.1.2. General Procedure for Population Generation

Seeds were pre-germinated on paper in a growth chamber set at 14-h photoperiod and day/night temperature regime of 30 °C/21 °C. One week after sowing, eight seedlings were transplanted into an 8-L pot containing field soil. The plants were thinned to four per pot 7 days later. Field soil, classified as Typic Albaqualf, was collected from the Centro Agropecuário da Palma (CAP/UFPel), located in Capão do Leão, RS, Brazil. The experiment was performed in a completely randomized design, with four replicates per cycle. The experimental unit consisted of one pot filled with soil containing four plants. G1 and G2 population production followed the same methodology used to produce G0 (Figure 1A).

Jungle rice plants were grown in a greenhouse under optimal growing conditions and were treated with a premix of imazapic + imazapyr (Kifix$^{TM}$) at the two- to three-leaf stage. All replicates were treated simultaneously using a $CO_2$ backpack sprayer calibrated to deliver a spray volume of 150 L·ha$^{-1}$. The recommended dose of Kifix$^{TM}$ herbicide for Echinochloa control in the irrigated rice field is 140 g·ha$^{-1}$ of commercial product (imazapic 175 g·Kg$^{-1}$ + imazapyr 525 g·Kg$^{-1}$). In this study, the low dose corresponded to 0.125× the recommended dose (Table 1). Plants were submitted to salt stress 24 h after herbicide application. Salt stress was implemented by irrigation with 120 mM NaCl solution; at a flood depth of 5 cm. Plants without both salt stress and herbicide stress was used as control. Salt stress treatment was imposed for 7 days, after which the remaining water was removed, and the designated plants were irrigated again with a saline solution for another 7 days. After a total of 14 days of salt treatment, the salty water was removed from all pots, and the plants were reflooded with regular water until maturity.

**Table 1.** Herbicide treatments used in experiments I and II on Echinochloa colona, at low herbicide use rates and salt stress. Both herbicides belong to WSSA Group 2, imidazolinone chemical family, which inhibit acetolactate synthase (ALS).

| Active Ingredient | Trade Name | Recommended Rate (g·ai·ha$^{-1}$) | Application Rate [a] (g·ai·ha$^{-1}$) |
|---|---|---|---|
| Imazapic + Imazapyr | Kifix$^{TM}$ | 24.5 + 73.5 | 3.06 + 9.19 |
| Imazethapyr | Newpath$^{TM}$ | 211 | 26.37 |

[a] Application rate: corresponding to 0.125× the recommended dose, based on preliminary experiments to allow survival and seed production (data not shown).

Before the flowering stage, the plants were isolated spatially to maintain purity. All seeds were collected at maturity and bulked for each treatment, producing the next generation. The selection cycle was repeated. During each selection cycle, a subpopulation of 16 plants from each saline water and regular water treatment, without herbicide, were grown to produce three generations of seeds without herbicide treatment.

2.1.3. Determination of Sensitivity Level to Imazapic + Imazapyr

While the parental population (G0) was being subjected to the first cycle of selection, a dose-response assay was conducted using the premix herbicide Kifix$^{TM}$. The dose-response bioassay was also conducted on G1 and G2 progenies; 30 days after each seed batch was harvested. The bioassay was conducted as follows.

Seeds were pre-germinated (previously described), and two seedlings were transplanted into each 0.7-L pot containing field soil. The plants were thinned to one per pot 7 days after transplanting. Upon reaching the two- to three-leaf stage, Kifix$^{TM}$ was applied at 0.0625×, 0.125×, 0.25×, 0.5×, 1.0×, and 2.0× the recommended dose, including a non-treated check, in three replicates. After 24 hours of herbicide application, the saline water treatment was imposed: without salt stress (water) and with salt stress (120 mM NaCl solution) for 14 d as described previously (Figure 1B).

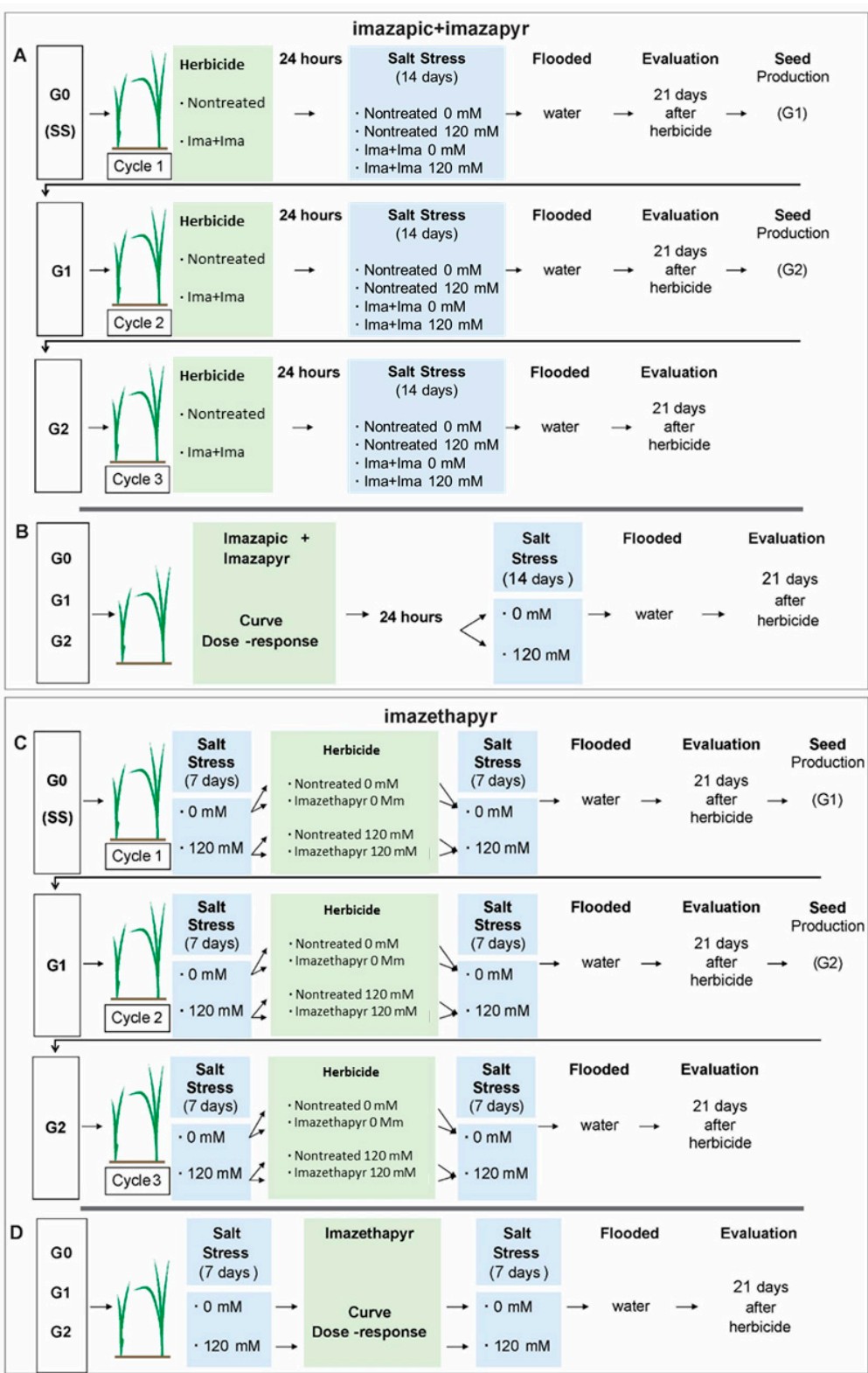

**Figure 1.** Schematic diagram of the progression of experiments on *Echinochloa colona*. (**A**) The low dose of imazapic + imazapyr and stress by salt. Parental population from the susceptible standard (G0) and selected (G1 and G2) progenies. Salt was applied at 120 mM. The sublethal herbicide dose was: nontreated check (no herbicide) and the formulated mixture of imazapic + imazapyr (at 0.125× the recommended dose). (**B**) Schematic diagram of the curve dose-response procedure on *E. colona* to imazapic + imazapyr. (**C**) The low dose of imazethapyr (0.125× the recommended dose) and stress by salt. (**D**) Schematic diagram of the curve dose-response procedure on *E. colona* to imazethapyr.



Three weeks after herbicide application, *E. colona* control was evaluated visually on a scale of 0% (no symptoms) to 100% (dead). Dose-response data were analyzed using the 'drc' package in R Software v.3.1.2 [38,39]. The three-parameter log-logistic model in Equation (1) was fitted to the data:

$$Y = d/1 + \exp[b(\log x - \log e)] \qquad (1)$$

where Y is the response (% control); d is the asymptotic value of Y at the upper limit; b is the slope of the curve around e ($ED_{50}$: the herbicide rate giving response halfway between d and the lower asymptotic limit, which was set to 0); and x is the herbicide rate. Susceptibility index (SI) was calculated as the ratio of the $ED_{50}$ with salt stress of each generation (G0, G1, and G2) divided by the $ED_{50}$ of the same generations without salt stress.

### 2.2. Experiment II—Effect of Salt Stress and Recurrent Selection With a Sublethal Dose of Imazethapyr on the Herbicide Sensitivity of an E. colona Population from Arkansas, USA

#### 2.2.1. Plant Material

This study was conducted using seeds of *E. colona* (referred to hereafter as G0), originally ECO-S, collected in 2011 from Prairie County, AR, USA. The seeds were threshed, cleaned, and stored at 4 °C until use. From this population, three successive selection cycles were implemented in 2018–2019 to produce three generations of seed (Parental (G0), G1, and G2 progenies) using the general procedure detailed below.

#### 2.2.2. General Procedure for Recurrent Selection

Seeds were planted into 50-cell trays filled with commercial potting soil (Sun Gro Horticulture Canada Ltd., Vancouver, BC, Canada). Approximately one week after planting, seedlings were transplanted into square pots (7.6 cm wide, 10.2 cm tall) containing a 1:3 mixture by volume of commercial potting soil and field soil (Captina silt loam-fine-silty, siliceous, active, mesic typic fragiudults). The experiment was performed in a completely randomized design, with six replicates. Each experimental unit was a pot containing one plant. The process was repeated to produce G1 and G2 (Figure 1C). The experiment was conducted in the greenhouse at the University of Arkansas, Fayetteville, AR, USA, with 14-h daylength under a day/night temperature regime of 30 °C/25 °C ± 5 °C. Plants were sub-irrigated and grown to two- to three-leaf stage when the salt and herbicide treatments were applied as described previously (Table 1). All replicates were treated simultaneously in a spray chamber at a pressure of 221 kPa and a volume of 187 L·ha$^{-1}$.

#### 2.2.3. Evaluation of Sensitivity to Imazethapyr

A dose-response experiment was conducted in the greenhouse at the University of Arkansas, Fayetteville, AR, USA, in January–February 2019, using G0, G1, and G2 seeds. Imazethapyr was applied at the same dose range (relative to the field dose) as the herbicides tested in experiment I (Figure 1D). A three-parameter log-logistic model was fitted to the data (Equation (1)).

### 3. Results

In general, the herbicide sensitivity profile was similar across experiments/herbicides. Treatments without salt stress showed no change in $ED_{50}$ as the generations progressed (Table 2 and Figure 2A–F), with values of 6.92 ($\pm$0.23), 6.92 ($\pm$0.34), and 6.92 ($\pm$0.22) g·ai·ha$^{-1}$ for imazapic in populations G0, G1, and G2, respectively. The $ED_{50}$ values for imazethapyr were 61.41 ($\pm$3.90), 61.09 ($\pm$2.30), 61.35 ($\pm$2.62) g·ai·ha$^{-1}$, in G0, G1, and G2, respectively. These average $ED_{50}$ values correspond to approximately 0.28 and 0.29× the recommended dose of imazapic and imazethapyr, respectively, demonstrating that the sensitivity of junglerice did not change after three cycles of low-dose herbicide selection without salt stress.

**Table 2.** Parameter estimates (b, d, ED$_{50}$, and TI) for the three-parameter log-logistic regression model fitted to the dose-response of *Echinochloa colona* to imazapic.

| | Log-Logistic Regression Estimates [a] | | | | |
|---|---|---|---|---|---|
| **Treatments [b]** | *B* | *D* | **ED$_{50}$** | *p*-**Value [c]** | **SI [d]** |
| **Imazapic** | | | **(g·ai·ha$^{-1}$)** | | |
| G0·0 mM | −3.96 (0.61) | 101.47 (2.18) | 6.92 (0.23) | <0.05 | - |
| G0·120 mM | −3.63 (0.39) | 100.78 (1.88) | 4.49 (0.16) | <0.05 | 0.65 |
| G1·0 mM | −4.06 (0.92) | 101.24 (3.21) | 6.92 (0.34) | <0.05 | - |
| G1·120 mM | −2.74 (0.46) | 99.36 (3.41) | 5.22 (0.35) | <0.05 | 0.75 |
| G2·0 mM | −4.06 (0.61) | 101.24 (2.14) | 6.92 (0.22) | <0.05 | - |
| G2·120 mM | −3.01 (0.35) | 102.62 (2.19) | 6.43 (0.24) | <0.05 | 0.93 |
| **Imazethapyr** | | | **(g·ai·ha$^{-1}$)** | | |
| G0·0 mM | −2.21 (0.25) | 103.41 (3.41) | 61.41 (3.90) | <0.05 | - |
| G0·120 mM | −1.78 (0.18) | 104.71 (3.03) | 37.25 (2.58) | <0.05 | 0.61 |
| G1·0 mM | −2.53 (0.21) | 102.72 (2.11) | 61.09 (2.30) | <0.05 | - |
| G1·120 mM | −2.15 (0.16) | 103.64 (1.92) | 43.03 (1.73) | <0.05 | 0.70 |
| G2·0 mM | −2.62 (0.27) | 102.74 (2.44) | 61.35 (2.62) | <0.05 | - |
| G2·120 mM | −2.20 (0.21) | 104.37 (2.47) | 53.38 (2.50) | <0.05 | 0.87 |

[a] Values are means and standard errors for parameter estimates in parentheses. [b] The numbers 0 and 120 mM were the salt treatments; G0 was the first cycle of selection and G1 and G2 were the subsequent generations. [c] *p*-value: comparing the difference between water and salt stress solution at the same cycle by the SI function in the "drc" package in R. v3.3.0. [d] SI: Susceptibility index was calculated as the ratio of the ED$_{50}$ with salt stress of each accession generation (G0, G1, and G2) divided by the ED$_{50}$ of the same generations without salt stress.

The plants subjected to salt stress (120 mM NaCl solution) were more susceptible to the herbicides in the first selection cycle. However, the ED$_{50}$ values increased in the subsequent selection cycle such that in G1, the susceptibility index (SI) was 0.75 and 0.70 for imazapic and imazethapyr, respectively; the SI was 0.93 and 0.87 for imazapic and imazethapyr, respectively, in G2 (Table 2 and Figure 2A–F). With imazapic plus salt stress, the ED$_{50}$ was 4.49 g·ai·ha$^{-1}$ for the parental population (G0). This increased to 5.22 and 6.43 kg·ai·ha$^{-1}$ in the G1 and G2 progenies, respectively. With imazethapyr plus salt stress treatments, the ED$_{50}$ values were 37.25, 43.03, and 53.38 g·ai·ha$^{-1}$ G0, G1, and G2, respectively, after three cycles of recurrent selection.

These results demonstrate that sensitivity to imidazolinone herbicides reduced as the generations evolved through recurrent selection with a sublethal dose of imidazolinone herbicides under salt stress, supporting the hypothesis that *E. colona* can gradually adapt to this combination of abiotic stress factors (Figure 3). In the beginning, salt stress increased the sensitivity of *E. colona* to imidazolinone herbicides, but after three cycles of recurrent selection, the offspring had almost wholly overcome the detrimental effect of salinity.

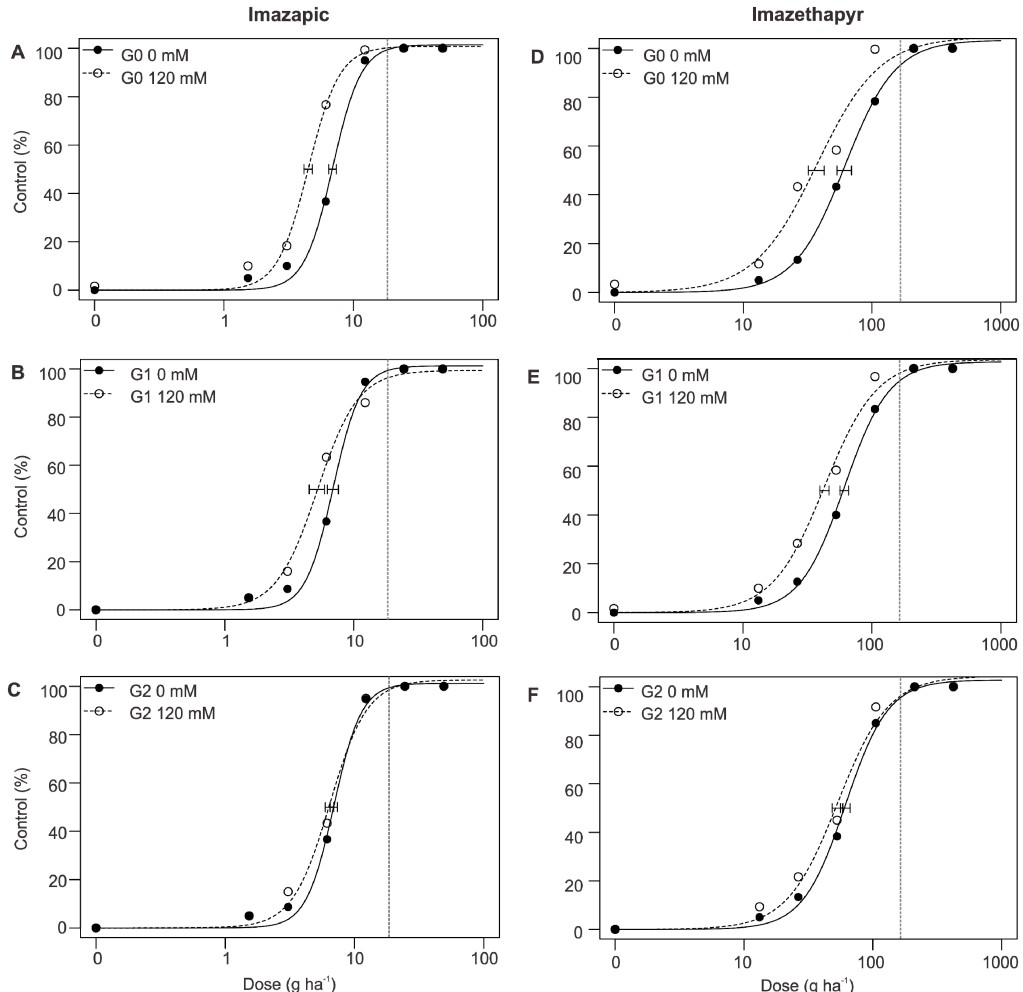

**Figure 2.** Nonlinear regression analysis of herbicide efficacy on *Echinochloa colona* as affected by cycles of recurrent exposure to salt stress and the sublethal dose of imidazolinone herbicides: response of *E. colona* from southern Brazil to imazapic (Kifix™, a formulated mixture of imazapic + imazapyr) in the parental population (G0) (**A**), G1 generation (**B**), and G2 generation (**C**). Also shown are the responses of *E. colona* from Arkansas, AR, USA to imazethapyr (Newpath™) of the parental population (G0) (**D**), G1 generation (**E**), and G2 generation (**F**). Circles (empty and filled) are the average of three replicates fitted with a three-parameter log-logistic model. Salt was applied at 120 mM. The vertical dotted gray lines indicate the recommended dose (g·ai·ha$^{-1}$).

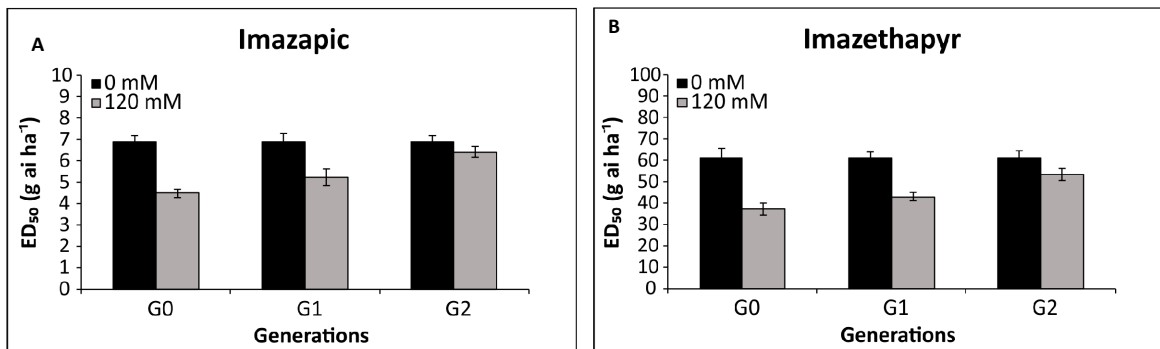

**Figure 3.** Results of ED$_{50}$ (herbicide efficacy) on *Echinochloa colona* as affected by cycles of recurrent exposure to salt stress and sublethal dose to the imidazolinones herbicides: application of a formulated mixture of imazapic + imazapyr (Kifix™) (A) and imazethapyr (B). Black and grey bars represent a salt concentration of 0 and 120 mM, respectively. Error bars indicate 95% confidence intervals.

## 4. Discussion

This study shows that the recurrent selection of *E. colona* with a sublethal dose of imazapic + imazapyr or imazethapyr combined with salt stress increases plant tolerance to these stress factors with transgenerational effect, resulting in reduced susceptibility to herbicide under salt stress. The formulated mixture of imazapic + imazapyr is a selective systemic herbicide of the imidazolinone chemical group, recommended for weedy rice control in the irrigated Clearfield® rice system [40]. It has the potential for leaching and persistence in the soil, which may cause contamination of underground water and phytotoxic effects to rotational crops up to two years after application in southern Brazil [41,42]. Imazethapyr, another herbicide from the imidazolinone chemical group, can be applied preemergence and postemergence for weed control in Clearfield® rice and needs adequate soil moisture to provide residual weed control [43]. The continuous use of these same herbicides over time in the same area can increase the carryover effect to sensitive rotational crops and increase the selection pressure on weeds, resulting in the evolution of herbicide-resistant genotypes [44,45].

In the plant environment system, there is a dynamic network of cause-and-effect interactions. Different herbicides and stresses, such as salinity, affect morphological, biochemical, physiological, and molecular attributes in crops and weeds [10,46–50]. Different responses at different biological levels have been studied in cultivated rice under salt stress [51–53]. However, there is little information about weed behavior under these conditions, and the weed response to a combination of herbicide and salinity stresses, which could be imprinted into the genome of succeeding generations. Markus et al. [36] and Rouse [37] proposed epigenetic regulation models that may be involved in herbicide resistance and adaptation to abiotic stresses. Additionally, Benedetti et al. [15,21] reported a reduction in sensitivity to certain herbicides in *E. colona* submitted to the recurrent selection of sublethal doses of herbicides and abiotic stresses, as drought and heat.

The response for each stress factor and its respective interactions is different for each plant and each stress and its combinations, and its effects are only observed when it impacts the physionomics of the plant, e.g., the growth and yield variables [54]. Ratogi et al. [54] observed a significant decrease on morphological parameter as in plant length, root length, shoot length, and leaf area with an increasing concentration of salt. In a study on the effect of different water salinity levels on the germination of imazamox-resistant and sensitive weedy rice and cultivated rice, Fogliato et al. [55] reported that salinity influenced not only the germination level of seeds, but also the time required to reach a same level of germination in both weedy rice and rice varieties. Additionally, imazamox-resistant rice and weedy rice had a similar germination speed, which was generally intermediate between that of Baldo (a conventional rice variety) and the sensitive weedy rice.

Considering that barnyardgrass and junglerice survived high salinity at 24 dS·m$^{-1}$ electrical conductivity, whereas rice did not, these species have a higher tolerance to salinity than rice [56]. Therefore, in areas that have saltwater intrusion problems, the availability of varieties, such as rice, that are tolerant to salinity is essential to maintain high yield [55,57]. Shrestha et al. [58] studied the effects of moisture and salinity stress on seed germination and salinity stress alone on growth and seed production on biotypes of *E. colona* resistant and susceptible to glyphosate. They observed that the glyphosate-resistant biotype was more tolerant than the glyphosate-susceptible biotype to salinity stress at germination, in aboveground biomass and during seed production, showing to be more competitive. Suggesting that there could be high genetic variability and phenotypic plasticity in *E. colona* populations, which indicates an advantage for weeds, particularly *Echinochloa* spp., it might have wide and rapid adaptability to contrasting environments [56,58,59].

Although the literature contains many reports on herbicide and salinity stress, the combined effect of these stresses, and the way it is being imprinted in the stress-response memory across generations, has not been studied before in weeds. This study presents preliminary results for future investigations of the mechanisms involved in weed adaptation to multiple abiotic stresses, such as the application of herbicide and salinity. Further

investigations are required to study how the combination of salt stress and herbicide application affects the soil–plant environment system and how the recurrent selection improves or reduces the species tolerance to stress.

## 5. Conclusions

The effective dose for 50% control of the population values increased from the parental population to the G2 generation after recurrent selection with a sublethal dose of imazapic + imazapyr and imazethapyr combined with salt stress. Thus, the results support the hypothesis of reduced susceptibility to chemical control by recurrent selection of *E. colona* submitted to a sublethal dose of imidazolinone herbicides combined with salt stress, with a transgenerational effect.

**Author Contributions:** Conceptualization, L.B., N.R.-B. and L.A.d.A.; data curation, L.B., V.R.G. and V.E.V.; formal analysis, L.B., V.R.G. and V.E.V.; funding acquisition, L.A.d.A. and N.R.-B.; investigation, L.B., P.C.-M. and N.R.-B.; methodology, L.B., N.R.-B., V.R.G., V.E.V., P.C.-M., G.M.S., E.R.C. and L.A.d.A.; project administration, N.R.-B. and L.A.d.A.; resources, N.R.-B. and L.A.d.A.; Supervision, G.M.S., E.R.C., L.A.d.A. and N.R.-B.; validation, L.B., V.R.G. and V.E.V.; visualization, L.B. and V.E.V.; writing—original draft, L.B.; writing—review and editing, N.R.-B., L.B., V.R.G., V.E.V., P.C.-M., G.M.S., E.R.C. and L.A.d.A. All authors have read and agreed to the published version of the manuscript.

**Funding:** This research received funding from: the Coordenação de Aperfeiçoamento de Pessoal de Nível Superior-Brasil (CAPES)-Finance Code 001; a BASF Corporation grant to N.R.-B.; The University of Arkansas (Hatch Project ARK02416), Fayetteville, USA; the research fellowship of L.A.A. by the Conselho Nacional de Desenvolvimento Científico e Tecnológico (CNPq-Proc.N. 310538/2015–7); and the student sandwich doctoral fellowship of L.B. was financed by the CNPq - Proc.N. 208443/2017–7 CNPq. This research made possible by the direct financing from the "Ciência Sem Fronteiras" public call MEC/MCTI/CAPES/CNPQ/FAPS-Visiting Researcher fellowship-PVE 2014 (Public Call number 401381/2014–5).

**Acknowledgments:** The authors thank the agencies that support this research and all the personal support from the Universidade Federal de Pelotas and the University of Arkansas.

**Conflicts of Interest:** The authors declare no conflict of interest.

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
