# Peer review of "Recurrent Selection with Low Herbicide Rates and Salt Stress Decrease Sensitivity of Echinochloa colona to Imidazolinone"

_agriculture, doi:10.3390/agriculture11030187_

Round 1
Reviewer 1 Report
Benedetti et al - recurrent selection paper review
In my view this paper provides evidence for recurrent selection of (at a phenomenological/phenotype level) a heritable reduction in the deleterious interaction for E. colona plants between herbicide application and salinity at time of treatment. It is not clear from the data provided whether recurrent herbicide selection plays any role in the effects observed (contrary to the authors' statements on their results), or whether there is any interaction between recurrent selection with saline conditions and recurrent selection with herbicide. From their description of the experiments, I suspect the authors have more control-group data available that they could use to elucidate these differences in mechanism, and which I would like to see included and discussed.
Detailed Comments:
Title
line 3: Consider adding botanical name after junglerice, to aid in searches.
Abstract
line 28: Spell out Echinochloa (I think this is the first appearance of the botanical name?)
Introduction
line 45-46: Perhaps extend these remarks to other key crops; the same problems occur in wheat, legumes, and other staples, as well as in key fibre crops.
line 48: (and elsewhere for other spp) I'm not sure of the journal's rules for botanical names, but if you haven't confirmed they don't want the full name with authority on first appearance, best do so.
line 49: I'd say TSR and NTSR are two groups of mechanisms; TSR is a type of mechanism but NTSR (as you imply later) includes several types of mechanisms within it.
line 55-56: It might improve readability for you to introduce recurrent selection at low doses more fully. Why are low doses investigated/important agronomically? What is recurrent selection and what does it reveal? There are hypothesised links between plant adaptation/adaptability to abiotic stress and rapid evolution of resistance to low-dose herbicide applications as a similar abiotic stress (i.e. that they may use similar genetic pathways), which would support your work and the reader's reasons for being interested in it.
line 62: 'Climate change impacts ON the ecosystem'
line 65: Remove The from 'The soil salinity'
line 75: Perhaps link water quality for herbicide use more explicitly with your previous points on adaptability (i.e. in saline soils where groundwater is used for herbicide application, the reduction in herbicide efficacy compounds with weed species' existing adaptability to abiotic stresses?)
Materials and Methods
line 115: I presume the saline-treated and untreated non-herbicide sub-populations were isolated from each other? It's not specified here.
equation 1: I found the use of e as a variable name potentially confusing. I'd consider it a reserved term in the same way that you wouldn't redefine pi as something other than the usual. (Also: the upper asymptotes from the graphs in Fig 2 appear to be above 100. Why wasn't the upper asymptote fixed at 100? I may not fully be understanding the mathematical method but >100 seems counter-intuitive.)
Results
Figure 2 and Table 2: The choice of comparing between salt/no-salt subpopulations within a generation, rather than between successive generations within salt/no-salt groups seems like an odd choice given the transgenerational essence of the research question.
Why is the p-value repeated for every sub-population? It's not as clear what your comparisons are in the table as it could be.
Table 2: I'd prefer to see something a bit more specific than <0.05 for p-values (is this a journal style thing?). Also, as there is no clear verbal explanation of the significance of the results, I'm left wondering whether you actually meant >0.05 (since it at least makes sense to be non-specific for non-significant results). But the SI values are at least different. So I'm not sure what your p-value results really are saying here.
Where are your control comparisons in these results? It appears to me from your description in figure 1 that you grew populations over each generation that were selected and not selected with herbicides, and selected and not selected with salt stress. But Fig 2 and table 2 only seem (as I read them) to include comparisons between three generations of the herbicide selected PLUS salt stress selected populations, tested for herbicide response with and without saline conditions. For example, in Fig 2C, there is no data for G2 plants from populations that had never been selected with salt stress but had been with herbicides, nor plants that had never been selected with herbicides but had been with salt stress. Am I correct in thinking this data exists but was not presented? (Arguably there is an indirect comparison between plants that had never been selected with either, being the G0 data in fig 2A/2D, but I don't think this provides all the data or analysis that we need here.)
ED50 increases in subsequent generations in the salt-stressed subpopulation but only for plants that are salt-stressed at time of treatment. It's not clear to me whether the selection is FOR tolerance of herbicide or of salt stress, though, nor whether the selection is BY herbicide application or salt stress, or both. Data on unsprayed plant growth might have addressed these questions but it appears you didn't collect that kind of data. It seems likely that the selection is for salt tolerance, since the main difference is between plants that are and are not subjected to salt stress at treatment time, but it's hard to say when there is no data shown for herbicide-selected but not salt-selected plants (nor the reverse, as far as I can tell). You demonstrated that the trait/s selected have some level of impact on herbicide performance over three generations, but not that herbicide performance degrades further than the 'normal' situation (non-saline water, non-resistant plants). So, it's not clear to me that you've demonstrated any interaction between low-dose herbicide stress and salt stress in changes in the populations over three generations. From your experiment description I think you have data available to make wider comparisons between different groups than the ones you're showing here. What was the difference in herbicide susceptibility in each generation between the salt-stressed plants with no herbicide history compared to salt-stressed plants with herbicide selection? What was the susceptibility of herbicide-selected but not salt-selected plants under salt stress compared to under no salt stress in G2? If you can add these data to your analysis, I would like to see them, to assess the individual effects of recurrent salt selection and recurrent herbicide selection.
Figure 3: How is this different from the data in table 2? It seems repetitive without adding information to me. If you want to keep it, add units to the ED50 and give the full herbicide mix in the title of 3A.
Discussion
218-220: As above, my view is that what you have shown is that salt tolerance increases, and that you've shown that through a likely/previously demonstrated connection between salt effects on plant growth and herbicide susceptibility. It's not clear that recurrent herbicide selection had anything to do with the changes you observed. And since the salt-stressed plants only approached the ED50 of the unstressed G0 plants, and the ED50 of the unstressed (at treatment) plants was not changed by the conditions in your experiment, I read the implication that selection for herbicide resistance, by the recurrent application of the herbicide, was in fact probably absent. I suspect (as you imply, but do not provide evidence for) the plants are growing more robustly in G1 and G2, and that made the herbicide more effective, as observed in many prior studies. But this doesn't necessarily imply a herbicide resistance mechanism or an interaction at a mechanism level.
line 247: I don't think it is true to say that those prior studies support your results, especially given they weren't all herbicide related. Your results demonstrate, as those other studies did, the rapid adaptibility of a weed species to an abiotic stress, and that this adaptability has implications for weed management. (The quoted figures in Serra et al are also not very convincing, given they almost entirely overlap between subgroups and I'm not sure that they were analysed as significantly different in the original paper.)
line 250: Reword 'and it is being imprinted' -- unclear.
Author Response
Dear Reviewer 1,
First of all, thank you very much for all your comments and for reviewing thoroughly the manuscript. We split the general comment to answer the questions point by point.
Reviewer 1 comments
Comment: Title line 3: Consider adding botanical name after junglerice, to aid in searches.
Response: We added the botanical name.
Comment: Abstract line 28: Spell out Echinochloa (I think this is the first appearance of the botanical name?)
Response: We added the full name.
Comment: Introduction line 45-46: Perhaps extend these remarks to other key crops; the same problems occur in wheat, legumes, and other staples, as well as in key fibre crops.
Response: We opt to describe only “grain yield” without focus for a specific crop.
Comment: Introduction line 48: (and elsewhere for other spp) I'm not sure of the journal's rules for botanical names, but if you haven't confirmed they don't want the full name with authority on first appearance, best do so.
Response: The species full name was added.
Comment: Introduction line 49: I'd say TSR and NTSR are two groups of mechanisms; TSR is a type of mechanism but NTSR (as you imply later) includes several types of mechanisms within it.
Response: We inserted the word “main”, “two main types”.
Comment: Introduction line 55-56: It might improve readability for you to introduce recurrent selection at low doses more fully. Why are low doses investigated/important agronomically? What is recurrent selection and what does it reveal? There are hypothesised links between plant adaptation/adaptability to abiotic stress and rapid evolution of resistance to low-dose herbicide applications as a similar abiotic stress (i.e. that they may use similar genetic pathways), which would support your work and the reader's reasons for being interested in it.
Response: We added in the text
Comment: Introduction line 62: 'Climate change impacts ON the ecosystem'
Response: We added in the text
Comment: Introduction line 65: Remove The from 'The soil salinity'
Response: We remove The from 'The soil salinity' in the text
Comment: Introduction line 75: Perhaps link water quality for herbicide use more explicitly with your previous points on adaptability (i.e. in saline soils where groundwater is used for herbicide application, the reduction in herbicide efficacy compounds with weed species' existing adaptability to abiotic stresses?)
Response: Our focus is not in the water quality for herbicide application, we aimed to study the effect of interaction between saline water in irrigation and plant stress caused by the herbicide and vice-versa.
Comment: Materials and Methods line 115: I presume the saline-treated and untreated non-herbicide sub-populations were isolated from each other? It's not specified here.
Response: We stated that the plants were kept separate spatially “Before the flowering stage, the plants were isolated spatially to maintain purity. All seeds were collected at maturity and bulked for each treatment, producing the next generation. The selection cycle was repeated. During each selection cycle, a subpopulation of 16 plants from each saline water and regular water treatment, without herbicide, were grown to produce three generations of seeds without herbicide treatment”
Comment: Materials and Methods equation 1: I found the use of e as a variable name potentially confusing. I'd consider it a reserved term in the same way that you wouldn't redefine pi as something other than the usual. (Also: the upper asymptotes from the graphs in Fig 2 appear to be above 100. Why wasn't the upper asymptote fixed at 100? I may not fully be understanding the mathematical method but >100 seems counter-intuitive.)
Response: The variable e is defined by the statistical program, and described in this case as ED50. Concerning the graphics in figure 2, the points are in the upper limit fixed in 100, the statistical program plot the curve to adjust the points and form the curve tendency.
Comment: Results Figure 2 and Table 2: The choice of comparing between salt/no-salt subpopulations within a generation, rather than between successive generations within salt/no-salt groups seems like an odd choice given the transgenerational essence of the research question. Why is the p-value repeated for every sub-population? It's not as clear what your comparisons are in the table as it could be.
Response: With these data the objective was to identify the salinity response in each generation. In this way, it was possible to assign a value to the SI by identifying the transgenerational effect of salinity and the herbicide and to represent the profile in each generation, without salt stress.
Comment: Results Table 2: I'd prefer to see something a bit more specific than <0.05 for p-values (is this a journal style thing?). Also, as there is no clear verbal explanation of the significance of the results, I'm left wondering whether you actually meant >0.05 (since it at least makes sense to be non-specific for non-significant results). But the SI values are at least different. So I'm not sure what your p-value results really are saying here.
Response: About p-value, the statistical program used indicates the significance of the data, in these cases, showing only that the value of p<0.05. In addition, based on other articles that use the same program and obtained significant responses, we used the same standard for displaying the results. For example, the p-value resulting from the statistical analysis was <2.2e-16 for all comparison between with and without salt in each generation, we chose to standardize and use the significant p-value <0.05.
Comment: Results Where are your control comparisons in these results? It appears to me from your description in figure 1 that you grew populations over each generation that were selected and not selected with herbicides, and selected and not selected with salt stress. But Fig 2 and table 2 only seem (as I read them) to include comparisons between three generations of the herbicide selected PLUS salt stress selected populations, tested for herbicide response with and without saline conditions. For example, in Fig 2C, there is no data for G2 plants from populations that had never been selected with salt stress but had been with herbicides, nor plants that had never been selected with herbicides but had been with salt stress. Am I correct in thinking this data exists but was not presented? (Arguably there is an indirect comparison between plants that had never been selected with either, being the G0 data in fig 2A/2D, but I don't think this provides all the data or analysis that we need here.)
Response: For all generations and graphs in Figure 2, recurrent selections of the herbicide without salt stress (empty/white circles) and with salt stress (full/black circles) were used to define the lethal dose of the herbicide in each generation and in each treatment (with and without salt stress). We performed a dose-response curve for each herbicide, using treatments without and with salt stress, in order to identify the ED50 in the original population. Thus, it could be possible to compare generations with the effect of salt stress on susceptibility to the herbicide.
Comment: Results ED50 increases in subsequent generations in the salt-stressed subpopulation but only for plants that are salt-stressed at time of treatment. It's not clear to me whether the selection is FOR tolerance of herbicide or of salt stress, though, nor whether the selection is BY herbicide application or salt stress, or both. Data on unsprayed plant growth might have addressed these questions but it appears you didn't collect that kind of data. It seems likely that the selection is for salt tolerance, since the main difference is between plants that are and are not subjected to salt stress at treatment time, but it's hard to say when there is no data shown for herbicide-selected but not salt-selected plants (nor the reverse, as far as I can tell). You demonstrated that the trait/s selected have some level of impact on herbicide performance over three generations, but not that herbicide performance degrades further than the 'normal' situation (non-saline water, non-resistant plants). So, it's not clear to me that you've demonstrated any interaction between low-dose herbicide stress and salt stress in changes in the populations over three generations. From your experiment description I think you have data available to make wider comparisons between different groups than the ones you're showing here. What was the difference in herbicide susceptibility in each generation between the salt-stressed plants with no herbicide history compared to salt-stressed plants with herbicide selection? What was the susceptibility of herbicide-selected but not salt-selected plants under salt stress compared to under no salt stress in G2? If you can add these data to your analysis, I would like to see them, to assess the individual effects of recurrent salt selection and recurrent herbicide selection.
Response: Both table 2 and figure 2 show the following data: Generations with and without salt, and also generations with and without herbicide, since to perform the dose response curve of each herbicide, the dose zero “0” was used as a nontreated check. In addition we demonstrate the effect of the factors alone (salt x herbicide x generations) and the interaction of these factors. For example: G0, without salt (0 mM), without herbicide (dose 0 of the curve), with herbicide (other doses) FIG 2 A (empty circules). G0, with salt (120 mM), without herbicide (dose 0 of the curve), with herbicide (other doses) FIG 2 A (full circules). For other generations, they follow the same exposure standard. Therefore, it is possible to identify the reduction in the susceptibility of the herbicide due to the saline treatment in G2.
Comment: Results Figure 3: How is this different from the data in table 2? It seems repetitive without adding information to me. If you want to keep it, add units to the ED50 and give the full herbicide mix in the title of 3A.
Response: We have added figure 3 in the text to facilitate the reader's visualization in identifying the treatments and the respective results. We do not include the full name of the herbicide in FIG 3 A, as it is described in table 1 of the section materials and methods that we use as a basis the recommended dose of imazapic which is 24.5 g ai ha– 1 of the formulated mixture. We include in figure 3 the ED50 unit as suggested.
Comment: Discussion 218-220: As above, my view is that what you have shown is that salt tolerance increases, and that you've shown that through a likely/previously demonstrated connection between salt effects on plant growth and herbicide susceptibility. It's not clear that recurrent herbicide selection had anything to do with the changes you observed. And since the salt-stressed plants only approached the ED50 of the unstressed G0 plants, and the ED50 of the unstressed (at treatment) plants was not changed by the conditions in your experiment, I read the implication that selection for herbicide resistance, by the recurrent application of the herbicide, was in fact probably absent. I suspect (as you imply, but do not provide evidence for) the plants are growing more robustly in G1 and G2, and that made the herbicide more effective, as observed in many prior studies. But this doesn't necessarily imply a herbicide resistance mechanism or an interaction at a mechanism level.
Response: The results showed that the effective dose for 50% control of the population (ED50) increased from the original population to G2, after the recurrent selection with a sublethal dose of imidazolinones combined with saline stress, which means that susceptibility was reduced by the combination of these abiotic stresses. Since more of the active ingredient is needed to control/kill 50% of the E. colona population submitted to the combination of these treatments. “…In the beginning, salt stress increased the sensitivity of E. colona to imidazolinone herbicides, but after three cycles of recurrent selection, the offspring had almost wholly overcome the detrimental effect of salinity…”
Comment: Discussion line 247: I don't think it is true to say that those prior studies support your results, especially given they weren't all herbicide related. Your results demonstrate, as those other studies did, the rapid adaptibility of a weed species to an abiotic stress, and that this adaptability has implications for weed management. (The quoted figures in Serra et al are also not very convincing, given they almost entirely overlap between subgroups and I'm not sure that they were analysed as significantly different in the original paper.)
Response: We changed the text, because our objective in citing these studies was to demonstrate that the species under study represents importance in the agricultural scenario, in addition to presenting greater tolerance to salt than rice crop, and that there are differences in several biological levels as to response of each individual. In addition, it presents different responses to factors such as herbicides and to different active ingredients.
Comment: Discussion line 250: Reword 'and it is being imprinted' -- unclear.
Response: The term "imprinted in the stress-response memory across generations" is commonly used in areas related to epigenetics, as it is a memory mechanism.

Reviewer 2 Report
Dear Authors, thank you for your interesting manuscript. Manuscript is interesting, but needs some clarifications:
What about statistical analysis? No description. Tests missing..(Tukey HSD or similar one), no significant differences between treatments - Fig. 3 - only show variance...
Fig. 2 - better split max 2 in one row - now it is not readable.
Current publications are missing (only few from 2020). I recommend adding:
10.1017/wsc.2017.79; 10.3390/agronomy9100658; 10.3109/07388551.2014.889080; 10.32615/ps.2019.169; 10.3390/su12176779; 10.1007/s00344-020-10235-9; doi:10.1614/0043-1745(2003)051[0610:PEOSIA]2.0.CO;2; 10.1614/WT-06-097.1
Add more information on the molecular and physiological flexibility of plants under salt stress (salinity). The discussion section needs to be revised. Arguments require a clearer and more accurate presentation. The understanding of physiological mechanisms is limited because it is limited to works that have a specific view and deliberately ignore alternatives, and do not represent a balanced view of the evidence.
Conclusions are very short and needs rewrite.
Author Response
Dear Reviewer 2,
First of all, thank you very much for all your comments and for reviewing thoroughly the manuscript. We split the general comment to answer the questions point by point.
Reviewer 2 comments
General comment from Reviewer 2:
Comment: Dear Authors, thank you for your interesting manuscript. Manuscript is interesting, but needs some clarifications:
Response: We agree that the manuscript have points to be enhanced and we took into account all the suggestions pointed.
Detailed comment from Reviewer 2:
Comment: What about statistical analysis? No description. Tests missing..(Tukey HSD or similar one), no significant differences between treatments - Fig. 3 - only show variance...
Response: 1- about statistical analysis: Section materials and methods: 2.1.3. Determination of sensitivity level to imazapic+imazapyr and 2.2.3. Evaluation of sensitivity to imazethapyr: “Dose-response data were analyzed using the ‘drc’ package in R Software v.3.1.2 Ritz; Seefeldt,. The three-parameter log-logistic model in Equation 1 was fitted to the data: Y = d/1 + exp[b(log x − log e)] Where Y is the response (% control); d is the asymptotic value of Y at the upper limit; b is the slope of the curve around e (ED50: the herbicide rate giving response halfway between d and the lower asymptotic limit, which was set to 0); and x is the herbicide rate. Susceptibility index (SI) was calculated as the ratio of the ED50 with salt stress of each generation (G0, G1, and G2) divided by the ED50 of the same generations without salt stress.” 2- about Fig 3.: We added figure 3 in the text to facilitate the reader's visualization in identifying the treatments and the respective results, but the values and the statistical analysis are the same from table 2 and figure 2 where we used the Dose-response data were analyzed using the ‘drc’ package in R Software and p-value results. Also, in Fig 3: Error bars indicate 95% confidence intervals.
Comment: Fig. 2 - better split max 2 in one row - now it is not readable.
Response: We changed
Comment: Current publications are missing (only few from 2020). I recommend adding:
10.1017/wsc.2017.79; 10.3390/agronomy9100658; 10.3109/07388551.2014.889080; 10.32615/ps.2019.169; 10.3390/su12176779;10.1007/s00344-020-10235-9; 10.1614/0043-1745(2003)051[0610:PEOSIA]2.0.CO;2; 10.1614/WT-06-097.1
Response: We edit the text and add some of them.
Comment: Add more information on the molecular and physiological flexibility of plants under salt stress (salinity). The discussion section needs to be revised. Arguments require a clearer and more accurate presentation. The understanding of physiological mechanisms is limited because it is limited to works that have a specific view and deliberately ignore alternatives, and do not represent a balanced view of the evidence.
Response: In this research, our objective was to study the response, in a weed species, of the interaction of salinity and imidazolinone herbicides in chemical control, since the information on the interaction of these factors, especially in weeds, is still incipient. Knowing the reduction in the demonstrated susceptibility, more studies should be carried out to clarify the interaction and the mechanisms involved in the responses, at the most varied biological levels. Just as we studied at the molecular level to drought and heat stress with herbicides in junglerice.
Comment: Conclusions are very short and needs rewrite.
Response: We rewrote.

Reviewer 3 Report
Minor comments to enhance the quality of the presentation:
Line 2-3: Please use upper case for each word in title subtitle and subheadings
Line 30-31: Please use ; to separate the keywords and delete . at the end
Line 30: Echinochloa colona (L.)
Line 48: please use the definition The acetolactate synthase (ALS) inhibitors
Line 81-82 and line 90 and throughout of the manuscript: Please use the capital letter for each word in sub-headings
Line 83: Plant Material
Line 115: treatment.

Author Response
Dear Reviewer 3,
First of all, thank you very much for all your comments and for reviewing thoroughly the manuscript. We split the general comment to answer the questions point by point.
Reviewer 3 comments
Comment: Line 2-3: Please use upper case for each word in title subtitle and subheadings
Response: We added.
Comment: Line 30-31: Please use ; to separate the keywords and delete . at the end
Response: We modified for ; and deleted . at the end.
Comment: Line 30: Echinochloa colona (L.)
Response: We added the full name at line 30.
Comment: Line 48: please use the definition The acetolactate synthase (ALS) inhibitors
Response: We added the definition “The acetolactate synthase (ALS) inhibitors” at line 48.
Comment: Line 81-82 and line 90 and throughout of the manuscript: Please use the capital letter for each word in sub-headings
Response: We added the capital letters in each word in sub-headings.
Comment: Line 83: Plant Material
Response: We changed to Plant Material.
Comment: Line 115: treatment.
Response: We changed to treatment.

Round 2
Reviewer 1 Report
I have no further issues with this MS
Reviewer 2 Report
Dear Authors,
now manuscript sounds better.
l.22 and l.32 - edit authornames of latin name... I recommended miss them...
caption 2.1.1 and 2.2.1 are same
still missing cleary description of used statistical methods